# Comprehensive review of statistical methods for analysing patient-reported outcomes (PROs) used as primary outcomes in randomised controlled trials (RCTs) published by the UK's *Health Technology Assessment* (HTA) journal (1997–2020)

Yirui Qian, Stephen J Walters, Richard Jacques, Laura Flight

School of Health and Related Research, The University of Sheffield, Sheffield, UK

**Correspondence to**
Professor Stephen J Walters;
s.j.walters@sheffield.ac.uk

## ABSTRACT

**Objectives** To identify how frequently patient-reported outcomes (PROs) are used as primary and/or secondary outcomes in randomised controlled trials (RCTs) and to summarise what statistical methods are used for the analysis of PROs.

**Design** Comprehensive review.

**Setting** RCTs funded and published by the United Kingdom's (UK) National Institute for Health Research (NIHR) Health Technology Assessment (HTA) Programme.

**Data sources and eligibility** HTA reports of RCTs published between January 1997 and December 2020 were reviewed.

**Data extraction** Information relating to PRO use and analysis methods was extracted.

**Primary and secondary outcome measures** The frequency of using PROs as primary and/or secondary outcomes; statistical methods that were used for the analysis of PROs as primary outcomes.

**Results** In this review, 37.6% (114/303) of trials used PROs as primary outcomes, and 82.8% (251/303) of trials used PROs as secondary outcomes from 303 NIHR HTA reports of RCTs. In the 114 RCTs where the PRO was the primary outcome, the most used PRO was the Short-Form 36 (8/114); the most popular methods for multivariable analysis were linear mixed model (45/114), linear regression (29/114) and analysis of covariance (13/114); logistic regression was applied for binary and ordinal outcomes in 14/114 trials; and the repeated measures analysis was used in 39/114 trials.

**Conclusion** The majority of trials used PROs as primary and/or secondary outcomes. Conventional methods such as linear regression are widely used, despite the potential violation of their assumptions. In recent years, there is an increasing trend of using complex models (eg, with mixed effects). Statistical methods developed to address these violations when analysing PROs, such as beta-binomial regression, are not routinely used in practice. Future research will focus on evaluating available statistical methods for the analysis of PROs.

## Strengths and limitations of this study

► Patient-reported outcomes (PROs) are increasingly used as clinical outcomes, and the use of appropriate statistical methods will ensure reliable estimates are produced to support decision making.

► This review summarises the up-to-date statistical methods that have been used for the primary analysis of PROs in 114 randomised controlled trials published by the UK's National Institute for Health Research (NIHR) *Health Technology Assessment* (HTA) journal between January 1997 and December 2020, and it is the largest comprehensive review of trials on the use of PROs and the corresponding statistical methods for the analysis of PROs to date.

► Most studies apply conventional statistical methods (eg, linear regression) for the analysis of PROs despite the potential violation of the model assumptions, and even though specialist statistical methods have been developed to solve this problem, these methods are rarely used in practice.

► There was a lack of explicit reporting of PROs in some trials. Although assumptions have been made based on the context where some required information was not explicitly stated, it is possible the extracted data were inconsistent with the researchers' intention.

► This study included only trials published in the NIHR HTA journal as the information relating to PROs in other journals are not reported in as much detail and the HTA journal intends to publish all NIHR-funded studies, regardless of their findings.

## INTRODUCTION

Patient-reported outcomes (PROs) are health or health-related outcomes of health interventions reported by patients themselves. PROs enable health researchers to measure,

analyse and compare clinical outcomes from the patient perspective and provide clinical effectiveness outcomes to support decision making. The descriptive and scoring system of a PRO can transform subjective descriptions of health to numerical scores in a range of dimensions,[1] and this transformation quantifies and allows the statistical analysis of these health-related outcomes.

Randomised controlled trials (RCTs) are regarded as the gold standard for evaluating the effectiveness of interventions.[2 3] The randomisation process in a well-conducted RCT can reduce the selection bias and allocation bias, and inform the causality of the treatment on responses.[4] These traits of RCTs can simplify the analysis of PRO data. However, the RCT methodology does not bypass the possible systematic error from sources such as invalid measurements (eg, poorly or mistakenly filled PRO forms, ceiling or floor effects of the PRO scales), publication bias or selective reporting of statistical analyses.

Despite these traits of RCTs, it can still be complex to analyse PRO data in RCT settings for the following reasons.

First, one PRO can generate multiple outcomes and can be reported in different forms—using different score types such as subscales or summary scores; generating quality-adjusted survivals (ie, the integration of the quality-of-life function over the observed period of survival, such as quality-adjusted life years); producing dichotomised outcomes (eg, proportion of responders with a PRO score above or below a particular prespecified value or cut-off); and producing ordinal outcomes using several cut-offs. The various forms can result in multiple endpoints, and potentially increase the complexity of analysing PRO data.

Second, PRO data are likely to be discrete, skewed and bounded (ie, with ceiling effect and floor effects).[5] When analysing PRO data using a general linear model (including t-test, analysis of variance (ANOVA), analysis of covariance (ANCOVA) and linear regression), there are a number of assumptions[6]:

1. The values of the outcome variable should have a Normal distribution for each value of the explanatory variable. This assumption means that the residuals are Normally distributed and should have a mean of 0.
2. Constant variance or homoscedasticity of the outcome variable at each value of the explanatory variable.
3. The relationship between the outcome variable and the explanatory variable should be linear.
4. Independent observations in the sample.

These assumptions such as Normality of residuals and linear relationship between the outcome independent variable and explanatory dependent variables are likely to be violated.[7 8]

Also, the application of statistical methods might vary according to PRO data and the aim of the statistical analysis, but these features (multidimensional, discrete, skewed and bounded) of PRO data may obscure the decision on what statistical methods need to be applied for the data analysis.

An inappropriate statistical analysis of PROs can result in unreliable estimands of clinical effectiveness and accordingly fail to provide accurate and robust results for decision making, with wider confidence intervals (CIs) or larger errors.[9 10] For example, patients may fail to receive an effective treatment because this treatment is falsely shown not to be clinically effective based on inaccurate estimands; vice versa, patients may receive treatment which may potentially harm their health when unreliable evidence supports the use of this treatment. Thus, applying appropriate statistical methods for the analysis of PROs is crucial to reduce biases of estimands, to accurately evaluate clinical effectiveness and to support healthcare decision making.

This study aimed to first identify how frequently PROs have been used as primary or secondary clinical outcomes in reports of RCTs published in the UK's National Institute for Health Research (NIHR) *Health Technology Assessment* (HTA) journal, and second, when the PRO was the primary outcome for trial, to summarise the statistical methods used to analyse the outcome.

## METHODS
### Definition of a PRO
There are several definitions of a PRO. For the purpose of this review, a PRO measure was defined as a questionnaire that measures health or a health-related outcome as a result of health interventions reported by patients themselves without any interpretation by clinicians or any other proxies. A PRO is an umbrella term for outcomes used to measure patients' perceptions of health-related quality of life (QoL), broader aspects of QoL, health status, satisfaction with the treatment and health conditions.[1 11–13]

### Trial identification
Reports of RCTs published in the UK's NIHR HTA journal between 1 January 1997 and 31 December 2020 that defined and reported a PRO as a clinical endpoint or outcome for the trial were systematically identified and reviewed. The HTA journal was chosen because, in comparison to major medical journals, the information related to the trial and the PROs are reported in more detail. Information related to the use of PROs included the frequency of using PROs as clinical outcomes; whether the PROs were used as primary and/or secondary outcomes; the characteristics of the PROs when they were used as primary outcomes; and when the PRO was the primary outcome, the statistical method used for the analysis of the PRO data. The identification of HTA reports of RCTs used the same search strategy as previous work.[14] The selection of trials with PROs was conducted by one reviewer (YQ). Three reviewers (SW, RJ and LF) conducted quality assurance checks on 30% of the included papers after the data extraction was completed, and disagreements were discussed to achieve consensus.

### Inclusion and exclusion criteria
The studies included in this review satisfied the following criteria: (1) RCTs where individual participants were

randomised, (2) trials with at least one PRO as the primary outcome and (3) trials with the statistical analysis conducted for the PRO. Studies excluded from this review included cluster RCTs as these have specific statistical issues, influenza trials as these rarely use PROs as clinical outcomes, adaptive or group sequential trials as these have different statistical issues that may influence the choice of analysis, and follow-on studies and pilot/feasibility studies.

PROs identified in this review can be well-established measures from previous studies with feasibility, reliability and validity already tested, or self-developed measures by researchers for use in the trial. For studies with measures which were not clearly defined as a PRO in the trial, various methods were taken to identify whether the measure was categorised as a PRO, including retrieving the cited paper that developed the measure, identifying signal words such as 'carers' and 'physicians' for rating or assessing patients' outcomes in the measure description, and referring to other papers that developed or applied the outcome measure. According to our definition of PROs, trials that recruited only patients, or trials that recruited both patients and proxies when patients were unable to complete PROs were included; trials that used only proxies as informants to complete the PROs were excluded in order to avoid the cases where clinicians respond to health outcome measures on the patients' behalf.

Trials using the product of PROs, such as a dichotomised outcome and quality-adjusted survival, were included. Trials that used PROs only as primary cost-effectiveness outcomes but not as clinical primary outcomes were excluded. Even if the statistical methods were proposed, trials that did not actually conduct the statistical analysis were excluded.

### Data extraction

The following information was extracted from the reports of the included trials:

1. Characteristics of the trials with PROs as primary outcomes, including the number of participants randomised and analysed, the baseline and postrandomisation assessment, the most frequently used PROs and special types of PROs (including patient satisfaction, preference-based and proxy-reported).
2. Statistical methods conducted for the primary analysis of the PROs, including the study population, the specific statistical methods, the adjustment for baseline score or other covariates, involvement of random effects, robust standard errors (SEs) and bootstrapping techniques, repeated measures analysis, and strategies for missing data.
3. The quality of reporting PROs, including whether there is a clear definition or justification of the primary outcomes or primary endpoints, statistical methods and covariates.

For the purpose of this review, the statistical methods were broadly categorised into two categories: 'univariable

methods' that do not adjust for any other covariates except the randomised group (eg, $t$-test, $\chi^2$ test and simple linear regression) and 'multivariable methods' that have one or more explanatory variables (eg, baseline score) in addition to the randomised group (eg, multiple linear regression). The multivariable methods were further classified according to the categories of generalized linear models (GzLM), including linear regression, ANCOVA, binary logistic regression, ordinal logistic regression, and their extensions for correlated responses such as models with coefficients estimated by generalized estimating equations (GEEs) and mixed effect models. Repeated measures analysis for PROs with more than one postbaseline assessments was classified into four categories: response feature analysis (ie, using summary measures, such as the area under the curve or postrandomisation mean score); generalized linear mixed models; GzLM with parameters estimated by GEE; and repeated measures ANOVA.[5 15]

### Patient and public involvement

Patients and the public were not involved in any way in this study.

### RESULTS

In total, 1356 reports were published by the HTA journal between 1 January 1997 and 31 December 2020, and 928 reports were excluded after screening the titles and abstracts. In the remaining 428 reports, 125 were excluded for various reasons (figure 1). In the 303 published individual RCTs, 37.6% (114/303) of trials used PROs as primary outcomes and 82.8% (251/303) of trials used PROs as secondary outcomes. Two trials with

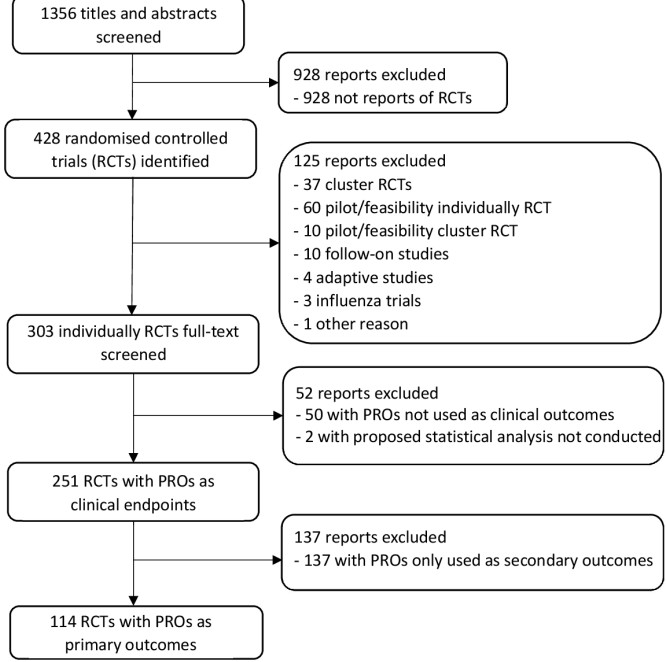

**Figure 1** Flow diagram for the inclusion and exclusion of trials published in the *Health Technology Assessment* journal from 1997 to 2020. PRO, patient-reported outcome.

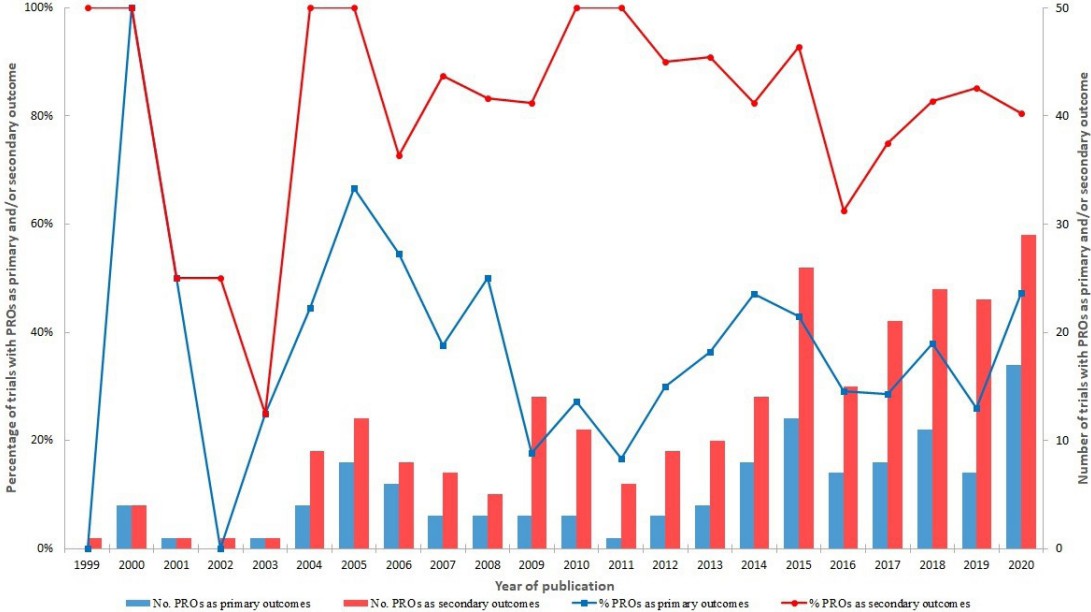

**Figure 2** Number and proportion of trials using patient-reported outcomes (PROs) as primary and/or secondary outcomes from 1999 to 2020.

PROs as primary outcomes were excluded as they were closed without conducting a statistical analysis of the data using the statistical methods that were proposed in the report.[16 17] It should be noted that the first RCT with a PRO as a clinical outcome was published in the HTA journal in 1999,[18] and the earlier reports published in the HTA journal were mainly systematic reviews.

All included trials that used PROs as primary outcomes also used PROs as secondary outcomes. The trend of using PROs as clinical outcomes in trials between 1999 and 2020 is shown in figure 2. Except for the earlier years (1999–2003) with a small number of studies, the proportion of trials with PROs used as secondary outcomes (the red curve) is approximately two times higher than the proportion of the trials with PROs as primary outcomes (the blue curve). Generally, there is an increase in using PROs as clinical outcomes in HTA trials.

### Trial characteristics

In total, 83.1% (61 715/74 298) of the participants randomised in the 114 trials were used for the primary analysis (table 1). The characteristics of these trials are summarised in table 2. The most common design was a two-arm parallel group trial. More than half of the trials were in either mental health (30/114) or musculoskeletal conditions (28/114). Most trials collected baseline

assessments (101/114) and more than one postbaseline assessments (107/114). The maximum number of postbaseline assessments was 24 in a trial on eczema management for children.[19]

Most trials (107/114) clearly defined the primary outcomes; the sample size calculation implied the primary outcomes for the trial was a PRO in six trials that did not explicitly specify the primary outcome, and one trial defined PROs as main outcome measures but used an alternative outcome for the sample size calculation.[20] Table 3 summarises the PROs used as primary outcomes in four or more included trials. The most popular PROs were mainly generic (ie, Short Form-36/Short Form 6-Dimension (SF-6D) and EuroQol-5 Dimension (EQ-5D)) and depression-specific (ie, Beck Depression Inventory, Hospital Anxiety and Depression Scale and Patient Health Questionnaire). Eight trials used more than one PRO as the primary outcomes, and 14 trials used non-PRO clinical outcomes as coprimary outcomes.

Preference-based PROs were used in six trials as primary outcomes, including five[21–25] that used the EQ-5D and one[26] that used the SF-6D. Seven trials used the quality-adjusted survival, including three[22 23 25] that used the EQ-5D and four[27–30] that used specific PROs for estimation. Patient satisfaction was used as the primary outcome

| Items | Mean | Median | SD | Min | Max | Total |
|---|---|---|---|---|---|---|
| Number of participants randomised | 652 | 480 | 928 | 85 | 8003 | 74 298 |
| Number of participants analysed* | 541 | 388 | 847 | 65 | 7677 | 61 715 |

**Table 1** Recruitment and retention of trial participants included from the 144 reports

*Number of participants analysed in the primary analysis of patient-reported outcomes; if multiple postbaseline assessments were used for the primary outcome, the number of participants analysed at the longest postbaseline timepoint was taken.

**Table 2** Trial design and assessment characteristics of the 114 trials included in the review

| Items | | No | % | Total |
|---|---|---|---|---|
| Trial design | Parallel group | 102 | 87.2 | 114 |
| | Factorial | 3 | 2.6 | |
| | Crossover | 0 | 0.0 | |
| | Other* | 9 | 7.7 | |
| Number of arms | 2 | 83 | 70.9 | 114 |
| | 3 | 21 | 17.9 | |
| | 4 | 5 | 4.3 | |
| | >4 | 5 | 4.3 | |
| Clinical area | Mental health | 30 | 25.6 | 114 |
| | Musculoskeletal | 28 | 23.9 | |
| | Obstetrics and gynaecology | 9 | 7.7 | |
| | Gastrointestinal | 7 | 6.0 | |
| | Respiratory | 5 | 4.3 | |
| | Stroke | 5 | 4.3 | |
| | Primary care | 4 | 3.4 | |
| | Cardiovascular | 4 | 3.4 | |
| | Cancer/oncology | 3 | 2.6 | |
| | Dermatology | 4 | 3.4 | |
| | Other† | 15 | 12.8 | |
| Number of trials with a baseline assessment of the patient-reported outcome | | 101 | 86.3 | 114 |
| Timing of primary outcome postbaseline assessments | <1 month | 7 | 6.0 | 114 |
| | 1–6 months | 28 | 23.9 | |
| | >6–18 months | 50 | 42.7 | |
| | >18 months | 27 | 23.1 | |
| | Missing‡ | 2 | 1.7 | |
| Number of postbaseline assessments | 1 | 5 | 4.3 | 114 |
| | 2 | 41 | 35.0 | |
| | 3 | 29 | 24.8 | |
| | 4 | 19 | 16.2 | |
| | >4 | 18 | 15.4 | |
| | Missing‡ | 2 | 1.7 | |

*Patient preference/Zelen's.
†Chronic fatigue, minor surgery, multiple sclerosis, neurosurgery, paediatric, sleep disorders, urology, vascular.
‡Two trials did not specify the timing and number of postbaseline assessments.

in two trials.[31 32] Proxies were recruited in six trials[19 33–37] that primarily aimed to collect PROs and only recruited proxies for patients who were unable to complete the PROs. Seven included trials used self-developed PROs as the primary or coprimary outcome. Most of these PROs had one item based on Likert Scale or Visual Analogue Scale (VAS), except for one trial[38] that specially developed a REFLUX questionnaire with 31 items to generate the QoL of patients with gastro-oesophageal

**Table 3** Most frequently used PROs as primary outcomes in the included trials

| PROs | Abbreviation | No | % | Reference |
|---|---|---|---|---|
| Short Form-36 | SF-36 | 8 | 7.0 | 79–82 |
| Short Form 6-Dimension | SF-6D | | | |
| Beck Depression Inventory | BDI | 7 | 6.1 | 83 |
| Hospital Anxiety and Depression Scale | HADS | 5 | 4.4 | 84 |
| EuroQol-5 Dimension | EQ-5D | 5 | 4.4 | 85–87 |
| Patient Health Questionnaire | PHQ | 5 | 4.4 | 88 |
| Oxford Shoulder Score | OSS | 4 | 3.5 | 89 |
| Other* | | 90 | 78.9 | |
| Total† | | 124 | 108.8 | |

*Only PROs that were used in four or more trials are listed separately.
†The total number of included trials is 114. Eight trials used more than one PRO as primary outcomes, including two trials that used three PROs and six trials that used two PROs as primary outcomes.
PRO, patient-reported outcome.

reflux disease. Health outcomes assessed by investigators were not included as primary outcomes, for example, four trials[39–41] using the Quality of Life Scale or Rankin Scale, that were completely assessed by investigators were excluded.

Different score types were used to summarise the PRO primary outcomes. Most trials (98/114) used one score type to report the primary outcome, and the rest of the trials used two types of scores. Summary scores were the most popular way to measure PROs (83/114), followed by subscales (30/114) and single items (16/114), including Likert Scale (11/114) and VAS (5/114).

### Statistical methods for the primary analysis of PROs

The majority of trials stated the proportion of missing PRO data (109/114), developed strategies to deal with missing data (99/114), imputed missing data using various methods such as mean imputation and last observation carried forward (89/114). In 40/114 studies, missing data were imputed as part of a sensitivity analysis to check the robustness of the primary analysis strategy which did not consider missing data.

Intention-to-treat (ITT) analysis (ie, analysis based on the randomised treatment assignment of all participants but not the actual treatment received),[42] including ITT with and without missing data imputation, was used in 111/114 trials, and 46 trials used other study populations such as per protocol analysis (ie, analysis based on the patients who completed the originally treatment

assigned), as treated or complier average causal effect analysis (ie, analysis of the treatment effect based on the subgroup that completed the originally treatment assigned) for the secondary or sensitivity analysis.

The statistical methods applied for primary analyses were clearly defined in 79/114 trials, and the use of univariable or multivariable methods for primary analyses were justified in 92/114 trials. Except for two trials that did not specify the timing and number of postrandomisation assessments,[43 44] 72/114 clearly defined the single timepoint used for the primary analysis (eg, x months postbaseline), and 40/114 used the repeated postbaseline outcomes for the primary analysis.

The statistical methods used for the primary analysis of PROs are shown in table 4. The linear mixed model (45/114) and linear regression (29/114) and ANCOVA (13/114) were the most popular methods. Among the 45 trials using linear mixed models for the primary analysis, 23 trials conducted a repeated measures analysis, and the rest did not consider the repeated postbaseline outcomes in the primary analysis. Three trials used GzLM with coefficients estimated by GEE for the longitudinal primary analysis, including one as an extension to ordinal logistic regression[45] and two as an extension to linear regression.[39 46] Only one trial with quality-adjusted survival used Cox regression for the primary analysis.[25]

Bootstrapped confidence intervals (CIs) were calculated after using a general linear model (including t-test, ANCOVA and linear regression) or linear mixed model in six trials[18 20 22 37 47 48] due to the skewness of PRO scores, and one additional trial did not conduct any other statistical analysis except calculating the bootstrapped CI for the treatment estimate.[49] Robust SEs can be used to estimate CIs and the calculation of test statistics (and associated P-values). Six trials[37 45 50–52] reported using robust SEs based on regression methods for the primary analysis, and three trials[46 53 54] reported using robust SEs for the longitudinal analysis in the non-primary analysis.

There were 106 trials that used multivariable methods, of which 98 clearly reported the covariates adjusted in the primary analysis. Among them, 85 trials adjusted for baseline score of the PRO, and 3 trials[55–57] modelled the change of PRO from baseline in the primary analysis. The use of random effects for the primary analysis of PROs was clearly specified in 47 trials: 44 used linear mixed models, and the other 3 trials used repeated measures ANOVA,[58] binary logistic mixed model[27] and ordinal logistic mixed model,[32] respectively. The most common random factors applied in the multivariable methods were therapists, centres (ie, hospital sites) and individual patients.

### Trend of using statistical methods for the primary analysis of PROs

Figure 3 shows the trend of using different statistical methods over the observed period. There is an increasing trend for using more complex models (mixed models and GLzM with GEE) in most recent years for the analysis of PROs. Linear regression, ANCOVA and repeated

**Table 4** Statistical methods used for the main or primary analysis of PROs that are the primary outcomes of the trials

| Statistical methods | N | % | Total |
|---|---|---|---|
| Univariable methods | | | 27 |
| t-test | 11 | 40.7 | |
| Unadjusted regression methods* | 7 | 25.9 | |
| $\chi^2$ test | 3 | 11.1 | |
| Wilcoxon rank-sum test (Mann-Whitney U test) | 4 | 14.8 | |
| Kruskal-Wallis test | 1 | 3.7 | |
| Log-rank test | 1 | 3.7 | |
| Multivariable methods† | | | 106 |
| Linear mixed model | 45 | 42.5 | |
| Linear regression | 29 | 27.4 | |
| ANCOVA | 13 | 12.3 | |
| Linear regression with GEE | 2 | 1.9 | |
| Binary logistic regression | 8 | 7.5 | |
| Binary logistic mixed model | 1 | 0.9 | |
| Ordinal logistic regression | 4 | 3.8 | |
| Ordinal logistic mixed model | 1 | 0.9 | |
| Repeated measures ANOVA | 6 | 5.7 | |
| Survival analysis | 1 | 0.9 | |
| Repeated measures analysis | | | 39 |
| Linear mixed model | 23 | 59.0 | |
| Response feature analysis‡ | 7 | 17.9 | |
| Repeated measures ANOVA | 6 | 15.4 | |
| GzLM with GEE | 3 | 7.7 | |

*In seven trials using unadjusted regression methods with no other covariates in the model besides the randomised group, three trials used linear regression; one used the linear mixed model; two used ordinal logistic regression; and one used binary logistic regression.
†106 trials used multivariable methods for the analysis of PROs, including 4 trials that used two different methods for the primary analysis of PROs.
‡In seven trials using response feature analysis (with quality-adjusted survivals) in primary analysis, four used linear regression or ANCOVA; two used linear mixed models; and one used survival analysis.
ANCOVA, analysis of covariance; ANOVA, analysis of variance; GEE, generalized estimating equation; GzLM, generalized linear model; PRO, patient-reported outcome.

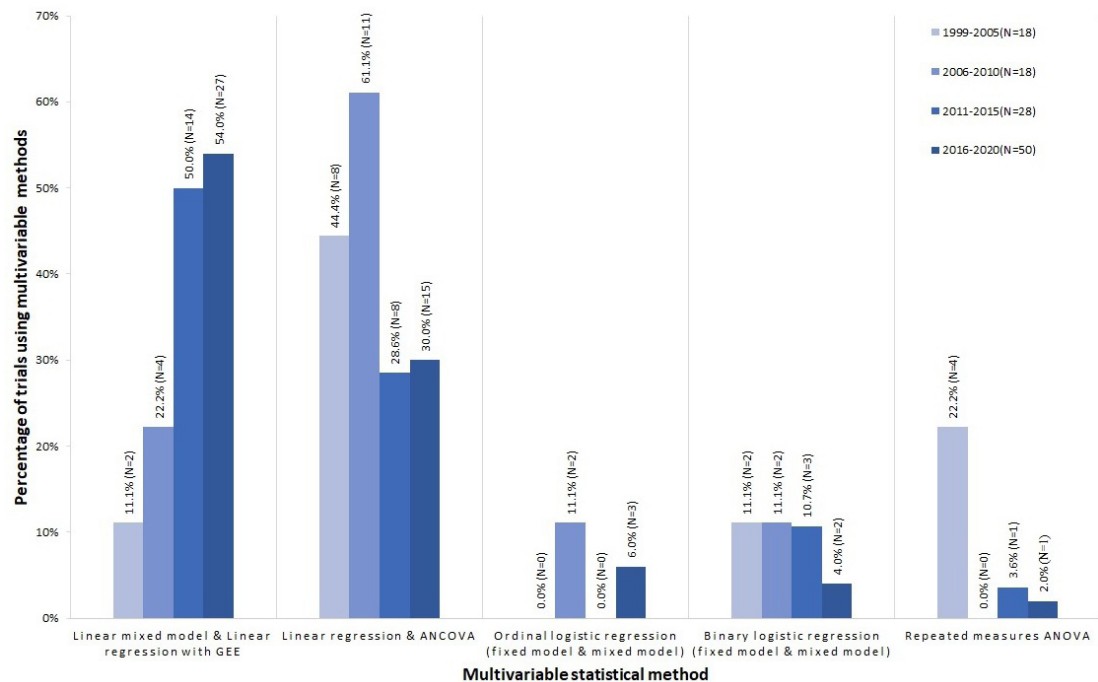

**Figure 3** Percentage of trials using multivariable methods for the primary analysis of PROs from 1999 to 2020 (N=xx) denotes the number of trials published in the specified period. As the survival analysis was used only in one trial, it is not shown in this graph. As this graph only summarised multivariable methods and one trial could use two or more multivariable methods for the primary analysis, the number of trials summarised in this graph may not equal the total number of included trials. ANCOVA, analysis of covariance; ANOVA, analysis of variance; GEE, generalized estimating equation.

measures ANOVA were the most popular regression methods used from 1999 to 2010, used in around 63% of trials on average. However, this popularity dropped to 28.6% during 2011–2015 and 30.0% during 2016–2020. In contrast, the proportion of trials using the linear mixed model and linear regression with GEE for correlated outcomes increased across the observation period, from 11.1% to 54.0%. While the use of binary logistic regression slightly decreased from 11.1% to 4.0% over time, the proportion of trials using ordinal logistic regression for the primary analysis remained small.

## DISCUSSION

This study has systematically conducted a review which summarised how frequently PROs have been used and what statistical methods have been applied for the primary analysis of PROs in RCTs published by the UK's NIHR HTA journal between 1997 and 2020. This review found that 82.8% (251/303) used PROs as primary or secondary outcome, and 37.6% (114/303) used PROs as primary outcomes. Though conventional methods (such as t-test, multiple linear regression and ANCOVA) have been widely used, there is a growing trend of using more complex methods such as linear mixed models (with both fixed and random effects) for the analysis of PROs over the observed period of this review.

The general linear model (including t-test, ANOVA, ANCOVA and linear regression) and linear mixed model were widely used over the past two decades, which could possibly result from the frequent use of continuous data type of response variables. The trend of using complex models could result from the increasing complexity of trial designs and from recommendations on using linear mixed models over repeated measures ANOVA for the longitudinal analysis of PROs.[5 59] There is a decrease in the use of binary logistic regression as the dichotomised outcome gathers less information from the PROs compared with other score types.[60]

There is a gap between the statistical methods developed by methodologists and statistical methods used for the analysis of PROs in trials. For example, Arostegui et al recommend the use of ordinal logistic regression with random effects model, beta-binomial regression or beta-logit-normal regression for continuous or ordinal PRO data after testing distributional assumptions.[59 61] However, only one of the five included trials that used ordinal logistic regression for the primary analysis considered random effects.[32] Neither beta-binomial regression nor beta-logit-normal regression were used by the 114 trials. Interestingly, the international standards for the analysis of PROs in cancer trials by the SISAQOL (Setting International Standards in Analysing Patient-Reported Outcomes and Quality of Life Endpoints Data) Consortium neither recommended the use of ordinal regression nor compared ordinal regression with other statistical methods.[62] The conventional methods are popular since the estimands produced by simple models can remain

robust regardless of the violation of model assumptions.[7] In addition, the difficulty in applying newly developed models in statistical software and the complexity to explain estimands of these models may make researchers more reluctant to use them.

PRO outcomes tend to generate data with discrete, skewed and bounded distributions that are not Normally distributed, and the assumptions for statistical methods such as the *t*-test, linear regression and ANCOVA may not be valid. However, Heeren and D'Agostino[63] have demonstrated the robustness of the two independent samples *t* test when applied to three-point, four-point and five-point ordinal scaled data using assigned scores, in sample sizes as small as 20 subjects per group. Sullivan and D'Agostino[64] have expanded this work to account for a covariate when the outcome is ordinal in nature. They again assign numeric scores to the distinct response categories and compare means between treatment groups adjusting for a covariate reflecting a baseline assessment measured on the same scale. Their simulation study shows that in the presence of three-point, four-point and five-point ordinal data and small sample sizes (as low as 20 per group), both ANCOVA and the two independent sample *t*-tests on difference scores are robust and produce actual significance levels close to the nominal significance levels.

Furthermore, statistical theory says that if the distribution of an outcome variable is Normal, so will be the distribution of the sample mean for that outcome variable. Much more importantly, even if the distribution of the outcome is not Normal, that of the sample mean will become closer to the Normal distribution as the sample size gets larger. This is a consequence of the Central Limit Theorem. The Normal distribution is strictly only the limiting form of the sampling distribution as the sample size increases to infinity, but it provides a remarkably good approximation to the sampling distribution even when the sample size is small, and the distribution of the outcome variable is far from Normal.[65] Thus, conventional statistical methods such as the *t*-test, linear regression/ANCOVA for analysing PROs are robust to the violation of assumptions for moderate to large sample sizes.[66]

To the best of our knowledge, this study is by far the largest review of trials (with 114 studies) published by the HTA journal which analysed the frequency of using PROs and the statistical methods for the analysis of PROs. The reviews by Pe *et al*[10] (breast cancer); Hamel *et al*[9] (lung cancer) and Fiteni *et al*[67] (lung cancer) had sample sizes of 66, 33, 27 articles, respectively. Compared with other reviews that concentrate on only one clinical area (ie, oncology),[9 10 67–70] this review summarised details in the frequency of using PROs and applying statistical methods in RCTs with a range of clinical areas.

It is noteworthy that the proportions of trials using PROs reported in this review represent the average rate of HTA trials focusing on different clinical areas, and when considering specific disease(s) or selecting different database(s), the proportions may vary. For example, Pe *et al*[10] identified 3/66 (5%) and 46/66 (70%) RCTs

of locally advanced and metastatic breast cancer using PROs as primary and secondary endpoints, respectively. Marandino *et al*[71] reviewed 446 cancer trials published in major journals between 2012 and 2016, and found that PRO or QoL was a primary endpoint in five trials (1.1%), a secondary endpoint in 195 trials (43.7%) and an exploratory endpoint in 36 trials (8.1%), while in the remaining 210 (47.1%), QoL was not listed at all among the study endpoints. Our review found that 3 of 18 cancer trials (17%) used PROs as primary outcomes, and 13/18 (72%) used PROs as secondary outcomes. Our results showed that PROs were more frequently used for health problems such as mental and musculoskeletal diseases.

This study has the following limitations. First, this review looked only at publicly funded trials in the UK, which may represent a limitation in terms of the generalisability of the findings. It is possible that statistical methods are used differently in industry-funded trials or in trials in other countries. However, as the NIHR HTA journal intends to publish all NIHR-funded projects, it has less publication bias compared with journals that publish only positive outcomes, and the information related to PROs in other journals is not reported in as much detail. The extracted statistical methods for the analysis of PROs from this review are consistent with those included from other similar reviews.[9 10 67–70] Second, there might be other appropriate methods for the analysis of PROs that were not included in this review. This review mainly analysed RCTs with PROs as primary outcomes because primary outcomes and the corresponding statistical methods were more explicitly reported.

Third, trials with PROs used only as cost-effectiveness outcomes were excluded. This is because the statistical strategies for clinical effectiveness and cost-effectiveness outcomes may vary, and cost-effectiveness analysis (CEA) produces both cost and clinical effectiveness outcomes. If PROs for CEA were included in this review, the proportion of included trials would increase, as there were some studies using EQ-5D for the primary CEA. It could be argued that the analysis of effectiveness estimated by PROs in CEA also requires appropriate statistical methods, but estimands calculated for a health economic analysis could be different from those for a clinical analysis as they hold different purposes to conduct these analyses. Therefore, we believe it is justified to make this exclusion.

We used a broad definition for a PRO, and a small number of trials (seven) used PROs that were specifically developed for the trial and were not validated in another external study. The inclusion of such non-validated instruments as primary outcomes should be discouraged and may have affected the results, although the characteristics of these PROs (Likert Scale or VAS) are similar to those of the PROs that have been formally validated. We believe that it is not unreasonable to assume that the statistical analysis of such outcomes would be similar to the analysis of validated PROs.

Another potential limitation is the large time window, 1997–2020, chosen for the review. This may introduce

some variability and potential heterogeneity in the trials included in the review, but on the positive side, it allows testing of time trends in the type of statistical methods used in the trials.

Last, the information related to PROs and statistical methods was not clearly reported in some trials. Although assumptions have been made based on the context where some required information was not explicitly stated, it is possible that the data extracted were inconsistent with researchers' intention. However, as the data have been extracted for all reports by one reviewer, there is consistency in the interpretation and assumptions made.

To produce explicit reports, it is recommended that researchers follow specific guidelines that can instruct the reporting of using PROs in RCT papers and protocols such as the CONSORT (Consolidated Standards of Reporting Trials) PRO (patient-reported outcomes) Extension,[13] SPIRIT (Standard Protocol Items: Recommendations for Interventional Trials) PRO (patient-reported outcome) Extension[72] and the standards for the analysis of PROs in cancer RCTs.[62] In addition, a clear classification of the terminology of the statistical methods is desired. It is a historical problem that the names of statistical methods are confusing (eg, general linear model vs GzLM), and multiple terms can be used to describe the same method; for example, the proportional odds model, ordered logit model and ordinal logistic regression refer to the same regression technique. Thus, researchers should be clear and cautious when describing the exact statistical method for the analysis.

In addition to the abovementioned issues, other obstacles are also worth attention when analysing PROs in RCT settings, which can be broadly classified into three domains: first, the statistics domain—inadequate understanding of basic statistical concepts,[73] incorrect procedures used to carry out statistical analysis and incorrect statistical inference (eg, secretly hypothesising after results are known)[74 75]; second, the PROs domain—application of invalid measurements,[76] lack of comparability among the results produced by various PROs[77]; and the third domain, reporting and publishing—publication bias and selective reporting of trials and their associated outcomes.[77 78]

In conclusion, the majority of trials funded by the NIHR HTA Programme used PROs as primary and/or secondary outcomes. Although there is an increasing trend of using complex models (eg, mixed effects), conventional methods such as linear regression remain widely used for the analysis of PROs, despite the potential violation of their assumptions. Statistical methods developed to address these violations when analysing PROs, such as beta-binomial regression, are not routinely used in practice. Various methods for the analysis of PROs have been identified from this review, but it is still unknown which methods are the most appropriate for the analysis of PRO data. Future research will focus on evaluating available statistical methods and make recommendations on using different methods for the analysis of PRO data.

**Contributors** All authors contributed to the study concept and study design. YQ and RJ contributed to the selection of data. YQ conducted the data extraction and data analysis and drafted the manuscript. SW, RJ and LF contributed to the quality assurance check of the data. All authors critically revised the manuscript and approved the final manuscript.

**Funding** YQ is sponsored jointly by the University of Sheffield and China Scholarship Council (grant number 201908890049). SW, RJ and LF received funding across various projects by National Institute for Health Research (NIHR). SW is an NIHR Senior Investigator supported by the NIHR (NF-SI-0617-10012) for this research project. The views expressed in this publication are those of the authors and not necessarily those of the NIHR, NHS or the UK Department of Health and Social Care. These organisations had no role in the study design; in the collection, analysis and interpretation of the data; in the writing of the report; or in the decision to submit the paper for publication.

**Competing interests** The PhD study of YQ is financially sponsored by the University of Sheffield and China Scholarship Council (grant number 201908890049). SW, RJ and LF received funding across various projects by the National Institute for Health Research (NIHR). SW is a senior investigator at NIHR (NF-SI-0617-10012) supported by the NIHR for this research project. The views expressed in this publication are those of the authors and not necessarily those of the NIHR, NHS or the UK Department of Health and Social Care. These organisations had no role in the study design; in the collection, analysis and interpretation of the data; in the writing of the report; or in the decision to submit the paper for publication.

**Patient and public involvement** Patients and/or the public were not involved in the design, conduct, reporting or dissemination plans of this research.

**Patient consent for publication** Not required.

**Ethics approval** The information extracted in this review is based on published HTA trials where ethics approvals were obtained by the original trial teams. This review does not involve recruiting new participants or analysing individual participants, and the original participants cannot be identified from this review.

**Provenance and peer review** Not commissioned; externally peer reviewed.

**Data availability statement** Data are available upon reasonable request. The information extracted in this review is based on published trials in the *Health Technology Assessment* (HTA) journal. The data extracted from the HTA Journal supporting the finding of this study is available on reasonable request from YQ at yqian21@sheffield.ac.uk.

**ORCID iDs**
Yirui Qian http://orcid.org/0000-0002-9276-5654
Stephen J Walters http://orcid.org/0000-0001-9000-8126
Richard Jacques http://orcid.org/0000-0001-6710-5403

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
