## [Reviewer comments · BMJ Open]

ARTICLE DETAILS

TITLE (PROVISIONAL)	A comprehensive review of statistical methods for analysing patient-reported outcomes (PROs) used as primary outcomes in randomised controlled trials (RCTs) published by the United Kingdom's Health Technology Assessment (HTA) Journal (1997-2020)
AUTHORS	Qian, Yirui; Walters, Stephen; Jacques, Richard; Flight, Laura

VERSION 1 – REVIEW

REVIEWER	Perrone, Francesco Istituto Nazionale Tumori IRCCS Fondazione Pascale Cura dei , Clinical Trials Unit
REVIEW RETURNED	Istituto Nazionale Tumori IRCCS Fondazione Pascale Cura dei , Clinical Trials Unit

GENERAL COMMENTS	The aim of this study are clearly presented and the background is solid. The paper is very well written and for a clinical reviewer like me (not formally a statistician) it was easy to read and interesting. Data interpretation and discussion is honest and sound. The stronger limitation of this study is the choice of the NIHR HTA Journal as the only source of clinical trials. This choice is explicitly justified by the Authors and while I acknowledge that it favours the pursuit of the objectives of the study, at the same time it represents a limitation in terms of generalizability of the findings. For example, the rate of trial using a PRO as primary endpoint (37.9%) or as a secondary (83.4%) is very high. This reviewer found that the corresponding figures in 446 cancer trials published in major journals between 2012 and 2016 were much lower: 1.1% and 52.9% (Annals of Oncology 29: 2288–2295, 2018 - doi:10.1093/annonc/mdy449). Might this be one of the biases deriving from choice of NIHR HTA Journal as the only source of data? Among PROs accepted for the study the Authors also report non validated “self-developed measures by researchers alongside trials”. I believe that the use of non validated instruments should be discouraged and I guess whether or not the inclusion of such non validated instruments affects the quality of the study. The very ample time window 1997-2020 is another bivalent characteristic of this study: on one side, it introduces variability and confusion, on the other side it allows to test time trends in the type of statistical methods used in the trials. On average, I find very interesting the time-trend analysis. However, a comment on the potential heterogeneity deriving from the large time-window might be added among the limitations.
---

	I would move the paragraph related to missing management on top of description of results, because this is a critical point to be addressed before any statistical analysis (from line 27-33 page 12 to around line 23 page 11).
--	--

REVIEWER	Raittio, Lauri Tampere University
REVIEW RETURNED	26-May-2021

GENERAL COMMENTS	Manuscript identifier: bmjopen-2021-051673 Title: A comprehensive review of statistical methods for analysing patient-reported outcomes (PROs) used as primary outcomes in randomised controlled trials (RCTs) published by the United Kingdom's Health Technology Assessment (HTA) Journal (1997-2020) The topic of the article is timely and important. I have few suggestions to merely discuss further in the article. Introduction, section 2: In addition to the three sentences, I would remind readers that the RCT methodology does not bypass the possible systematic error from sources such as invalid measurements (e.g. poorly or mistakenly filled PRO forms, ceiling or floor effects of the PRO scales), publication bias or selective reporting of statistical analyses. Introduction, section 3: In addition to the remind of the Normality assumption, for linear regression more important assumptions are: measurements are valid (see above), the relationship between measurement (PRO scores) and ones objective (how well patients are doing) is truly linear, and thirdly the dependent variables are additive in relation to the independent variable. Also, you could provide reference to some article that corrects some of the common misunderstandings about the normality assumption, e.g. [1]. Methods: Nothing to add, clearly written. Results, section 3: The most trials measured and reported the baseline assessment of the PRO scale but you haven't reported how many of the RCTs adjusted for this baseline score in the statistical analyses. Often the baseline and post-baseline assessments are correlated within individuals and not taking this into account (in my experience, quite common among small RCTs) increases the SEs of the estimands. See e.g. [2,3,12,13,4–11]. I'd suggest you report how often the baseline score of the PRO was adjusted in the sample of 114 trials. Also, the presence of covariate adjustment would be interesting to see among the sample, but it could enlarge the article too big, so make your own choice about that. Discussion: Overall the discussion section is balanced, although I'd prefer to see more educational or more elaboration scope in it. The current situation among medical RCTs regarding statistical analyses is suboptimal in many aspects: non-publication bias [14,15], selective reporting [16], secretly HARKing [17], poor conduct and analysis [18], overconfidence about responder analyses [19–22], poor comparability of the results due to endless amount of PRO scales [23–25], and lastly, inadequate understanding of the basic statistics concepts among medical researchers and other readers (clinicians) [26–31] just to name a few examples with overlapping themes with this study. Briefly, I would add to the discussion section some content on obstacles
--

about PROs and statistical analyses in the current atmosphere of medical research.

References:

1 Lumley T, Diehr P, Emerson S, et al. The Importance of the Normality Assumption in Large Public Health Data Sets. *Annu Rev Public Health* 2002;23:151–69.

doi:10.1146/annurev.publhealth.23.100901.140546

2 Vickers AJ. The use of percentage change from baseline as an outcome in a controlled trial is statistically inefficient: A simulation study. *BMC Med Res Methodol* 2001;1:1–4. doi:10.1186/1471-2288-1-6

3 Vickers AJ, Altman DG. Statistics notes: Analysing controlled trials with baseline and follow up measurements. *BMJ* 2001;323:1123–4. <http://www.ncbi.nlm.nih.gov/pubmed/11701584> (accessed 19 Jan 2019).

4 Clifton L, Clifton DA. The correlation between baseline score and post-intervention score, and its implications for statistical analysis. *Trials* 2019;20:43. doi:10.1186/s13063-018-3108-3

5 Ciolino J, Zhao W, Martin R, et al. Quantifying the cost in power of ignoring continuous covariate imbalances in clinical trial randomization. *Contemp Clin Trials* 2011;32:250–9.

doi:10.1016/j.cct.2010.11.005

6 Ciolino JD, Martin RH, Zhao W, et al. Measuring continuous baseline covariate imbalances in clinical trial data. *Stat Methods Med Res* 2015;24:255–72. doi:10.1177/0962280211416038

7 Ciolino JD, Palac HL, Yang A, et al. Ideal vs. real: A systematic review on handling covariates in randomized controlled trials. *BMC Med. Res. Methodol.* 2019;19:1–11. doi:10.1186/s12874-019-0787-8

8 Ciolino JD, Martin RH, Zhao W, et al. Covariate imbalance and adjustment for logistic regression analysis of clinical trial data. *J Biopharm Stat* 2013;23:1383–402.

doi:10.1080/10543406.2013.834912

9 Kahan BC, Morris TP. Reporting and analysis of trials using stratified randomisation in leading medical journals: review and reanalysis. *BMJ* 2012;345:e5840–e5840. doi:10.1136/bmj.e5840

10 Kahan BC, Jairath V, Doré CJ, et al. The risks and rewards of covariate adjustment in randomized trials: an assessment of 12 outcomes from 8 studies. *Trials* 2014;15:139. doi:10.1186/1745-6215-15-139

11 Hernández A V., Eijkemans MJC, Steyerberg EW. Randomized controlled trials with time-to-event outcomes: How much does prespecified covariate adjustment increase power? *Ann Epidemiol* 2006;16:41–8. doi:10.1016/j.annepidem.2005.09.007

12 Hernández A V., Steyerberg EW, Habbema JDF. Covariate adjustment in randomized controlled trials with dichotomous outcomes increases statistical power and reduces sample size requirements. *J Clin Epidemiol* 2004;57:454–60.

doi:10.1016/j.jclinepi.2003.09.014

13 Saquib N, Saquib J, Ioannidis JPA. Practices and impact of primary outcome adjustment in randomized controlled trials: Meta-epidemiologic study. *BMJ* 2013;347. doi:10.1136/bmj.f4313

14 Chan AW, Song F, Vickers A, et al. Increasing value and reducing waste: Addressing inaccessible research. *Lancet.* 2014;383:257–66. doi:10.1016/S0140-6736(13)62296-5

15 Song F, Parekh S, Hooper L, et al. Dissemination and publication of research findings: An updated review of related biases. *Health Technol Assess (Rockv)* 2010;14:1–220.

doi:10.3310/hta14080

	16 Goldacre B, Drysdale H, Dale A, et al. COMPare: A prospective cohort study correcting and monitoring 58 misreported trials in real time. Trials 2019;20. doi:10.1186/s13063-019-3173-2 17 Hollenbeck JR, Wright PM. Harking, Sharking, and Tharking. J Manage 2017;43:5–18. doi:10.1177/0149206316679487 18 Ioannidis JPA, Greenland S, Hlatky MA, et al. Increasing value and reducing waste in research design, conduct, and analysis. Lancet. 2014;383:166–75. doi:10.1016/S0140-6736(13)62227-8 19 Cortés J, González JA, Medina MN, et al. Does evidence support the high expectations placed in precision medicine? A bibliographic review. F1000Research 2018;7:30. doi:10.12688/f1000research.13490.5 20 Senn S. Seven myths of randomisation in clinical trials. Stat Med 2013;32:1439–50. doi:10.1002/sim.5713 21 Senn SJ. Statistical pitfalls of personalized medicine. Nature Published Online First: 2018. https://www.nature.com/articles/d41586-018-07535-2 22 Senn S. Individual Therapy: New Dawn or False Dawn? Drug Inf J 2001;35:1479–94. doi:10.1177/009286150103500443 23 Miyar J, Adams CE. Content and quality of 10 000 controlled trials in schizophrenia over 60 years. Schizophr Bull 2013;39:226–9. doi:10.1093/schbul/sbr140 24 Wuytack F, Smith V, Clarke M, et al. Towards core outcome set (COS) development: A follow-up descriptive survey of outcomes in Cochrane reviews. Syst Rev 2015;4. doi:10.1186/s13643-015-0060-0 25 Yordanov Y, Dechartres A, Atal I, et al. Avoidable waste of research related to outcome planning and reporting in clinical trials. BMC Med 2018;16:87. doi:10.1186/s12916-018-1083-x 26 Localio AR, Stack CB, Meibohm AR, et al. Inappropriate Statistical Analysis and Reporting in Medical Research: Perverse Incentives and Institutional Solutions. Ann Intern Med 2018;169:577. doi:10.7326/M18-2516 27 Goodman SN. Why is Getting Rid of P -Values So Hard? Musings on Science and Statistics. Am Stat 2019;73:26–30. doi:10.1080/00031305.2018.1558111 28 Gigerenzer G. Statistical Rituals: The Replication Delusion and How We Got There. Adv Methods Pract Psychol Sci 2018;1:198–218. doi:10.1177/2515245918771329 29 Gigerenzer G, Gaissmaier W, Kurz-Milcke E, et al. Helping doctors and patients make sense of health statistics. Psychol Sci Public Interes Suppl 2007;8:53–96. doi:10.1111/j.1539-6053.2008.00033.x 30 Greenland S. Null misinterpretation in statistical testing and its impact on health risk assessment. Prev Med (Baltim) 2011;53:225–8. doi:10.1016/J.YPMED.2011.08.010 31 Greenland S, Senn SJ, Rothman KJ, et al. Statistical tests, P values, confidence intervals, and power: a guide to misinterpretations. Eur J Epidemiol 2016;31:337–50. doi:10.1007/s10654-016-0149-3
--	--

VERSION 1 – AUTHOR RESPONSE

Comments from Reviewer 1

Comment 1: The aim of this study are clearly presented and the background is solid. The paper is very well written and for a clinical reviewer like me (not formally a statistician) it was easy to read and interesting. Data interpretation and discussion is honest and sound.

Response: Thanks for your appreciation.

Comment 2: The stronger limitation of this study is the choice of the NIHR HTA Journal as the only source of clinical trials. This choice is explicitly justified by the Authors and while I acknowledge that it favours the pursuit of the objectives of the study, at the same time it represents a limitation in terms of generalizability of the findings. For example, the rate of trial using a PRO as primary endpoint (37.9%) or as a secondary (83.4%) is very high. This reviewer found that the corresponding figures in 446 cancer trials published in major journals between 2012 and 2016 were much lower: 1.1% and 52.9% (Annals of Oncology 29: 2288–2295, 2018 - doi:10.1093/annonc/mdy449). Might this be one of the biases deriving from choice of NIHR HTA Journal as the only source of data?

Response: Thank you for pointing this out and referencing the journal article. We agree that the proportion of trials with PROs as clinical endpoints may vary when looking at trials that focused on different clinical areas and published in different journals. As the trials published in the HTA journal cover various clinical areas, we believe it is not sensible to extrapolate the derived proportions (37.6% & 82.8%) to a specific clinical area.

We have added the following paragraph to discuss this point. [page 16 line 45 to page 17 line 5 (Discussion: paragraph 7)].

'It is noteworthy that the proportions of trials using PROs reported in this review represents the average rate of HTA trials focusing on different clinical areas, and when considering specific disease(s) or selecting different database(s), the proportions may vary. For example, Pe et al[10] identified 3/66 (5%) and 46/66 (70%) randomised controlled trials of locally advanced and metastatic breast cancer using PROs as primary and secondary endpoints respectively. Marandino et al[82] reviewed 446 cancer trials published in major journals between 2012 and 2016, and found that PRO or QoL was a primary end point in five trials (1.1%), a secondary end point in 195 trials (43.7%), an exploratory endpoint in 36 trials (8.1%), while in the remaining 210 (47.1%) QoL was not listed at all among study end points. Our review found that three of eighteen cancer trials (17%) used PROs as primary outcomes, and 13/18 (72%) used PROs as secondary outcomes. Our results showed that PROs were more frequently used for health problems such as mental and musculoskeletal disease.'

Comment 3: Among PROs accepted for the study the Authors also report non validated “self-developed measures by researchers alongside trials”. I believe that the use of non validated instruments should be discouraged and I guess whether or not the inclusion of such non validated instruments affects the quality of the study.

Response: We agree that the non-validated self-developed PROs should not be recommended to use. According to our definition, PROs that report health outcomes or health-related outcomes from a patient perspective should be included in this study. We believe it is reasonable to count in all types of PROs even they were developed alongside the trial since they were PROs adapted in the trials and ignoring them would bias the outcome from our definition. Instead of excluding reports with self-developed PROs, we have added the following sentence briefly describing these trials. [page 11 line 11-18 (Results - Trial characteristics: paragraph 3)].

'Seven of the included trials used self-developed PROs as primary or coprimary endpoint. Most of these PROs had one item based on Likert scale or visual analogue scale (VAS), except for one trial[37] that specially developed a REFLUX questionnaire with 31 items to generate the QoL of patients with gastro-oesophageal reflux disease.'

In discussion, we have also added a paragraph to discuss this issue. [page 17 line 40-50 (Discussion paragraph 10)]

'We used a broad definition for a PRO and a small number of trials (seven) used PROs that were specifically developed for the trial and were not validated in another external study. The inclusion of such non-validated instruments as primary outcomes should be discouraged, and may have affected the results, although the characteristics of these PROs (Likert or VAS) are similar to those of the PROs that have been formally validated. We believe that it is not unreasonable to assume that the statistical analysis of such outcomes would be similar to the analysis of validated PROs.'

Comment 4: The very ample time window 1997-2020 is another bivalent characteristic of this study: on one side, it introduces variability and confusion, on the other side it allows to test time trends in the type of statistical methods used in the trials. On average, I find very interesting the time-trend analysis. However, a comment on the potential heterogeneity deriving from the large time-window might be added among the limitations.

Response: Thanks for this comment. We selected the time window as we would like to include as many qualified trials as possible. We have added a few sentences discussing this point. [page 17 line 52-57 (Discussion: paragraph 11)]

'Another potential limitation is the large time window, 1997-2020, chosen for the review. This may introduce some variability and potential heterogeneity in the trials included in the review, but on the positive side it allows to test time trends in the type of statistical methods used in the trials.'

Comment 5: I would move the paragraph related to missing management on top of description of results, because this is a critical point to be addressed before any statistical analysis (from line 27-33 page 12 to around line 23 page 11).

Response: Thanks for this suggestion. We have made this change accordingly.

Comments from Reviewer 2

The topic of the article is timely and important.

I have few suggestions to merely discuss further in the article.

Comment 6: Introduction, section 2: In addition to the three sentences, I would remind readers that the RCT methodology does not bypass the possible systematic error from sources such as invalid measurements (e.g. poorly or mistakenly filled PRO forms, ceiling or floor effects of the PRO scales), publication bias or selective reporting of statistical analyses.

Response: Thanks for this suggestion. We have revised the second paragraph of the introduction accordingly. [page 5 line 17-27]

'Randomised controlled trials (RCTs) are regarded as the gold standard for evaluating the effectiveness of interventions.[2,3] The randomisation process in a well-conducted RCT can reduce the selection bias and allocation bias, and inform the causality of the treatment on responses.[4] These traits of RCTs can simplify the analysis of PRO data. But the RCT methodology does not bypass the possible systematic error from sources such as invalid measurements (e.g. poorly or mistakenly filled PRO forms, ceiling or floor effects of the PRO scales), publication bias or selective reporting of statistical analyses.'

Comment 7: Introduction, section 3: In addition to the remind of the Normality assumption, for linear regression more important assumptions are: measurements are valid (see above), the relationship between measurement (PRO scores) and ones objective (how well patients are doing) is truly linear, and thirdly the dependent variables are additive in relation to the independent variable. Also, you could provide reference to some article that corrects some of the common misunderstandings about the normality assumption, e.g. [1].

Response: Thanks for this nice summary. We have revised this paragraph and added the model assumptions in the introduction. [page 5 line 44 to page 6 line 15 (Introduction: paragraph 4 and 5)]

'Second, PRO data is likely to be discrete, skewed, and bounded (i.e. with ceiling effect and floor effects).[5] When analysing PRO data using a general linear model (including t-test, ANOVA, ANCOVA and linear regression), there are a number of assumptions[6]:

1. The values of the outcome variable should have a Normal distribution for each value of the explanatory variable. This assumption means that the residuals are Normally distributed and should have a mean of zero;
2. Constant variance or homoscedasticity of the outcome variable at each value of the explanatory variable;
3. The relationship between the outcome variable and the explanatory variable should be linear;
4. Independent observations in the sample.

These assumptions such as Normality of residuals and linear relationship between outcome independent variable and explanatory dependent variables are likely to be violated.[7,8]

Also, the application of statistical methods might vary according to PRO data and the aim of statistical analysis, but these features (multidimensional, discrete, skewed, and bounded) of PRO data may obscure the decision on what statistical methods need to be applied for the data analysis.'

Methods: Nothing to add, clearly written.

Comment 8: Results, section 3: The most trials measured and reported the baseline assessment of the PRO scale but you haven't reported how many of the RCTs adjusted for this baseline score in the statistical analyses. Often the baseline and postbaseline assessments are correlated within individuals and not taking this into account (in my experience, quite common among small RCTs) increases the SEs of the estimands. See e.g. [2,3,12,13,4–11]. I'd suggest you report how often the baseline score of the PRO was adjusted in the sample of 114 trials. Also, the presence of covariate adjustment would be interesting to see among the sample, but it could enlarge the article too big, so make your own choice about that.

Response: This information about the number of trials adjusted for the baseline score in the statistical analysis is available on page 13 line 5-9 (Results - Statistical methods for the primary analysis of PROs: paragraph 6). Please note that the sentence is rephrased in the revised copy.

'Baseline values of the PRO were used to adjust analyses in 85 trials. Among them, 85 trials adjusted for baseline score of the PRO'.

We decide not to extend the article by adding more details of the covariates in the sample, and we keep the initial description for the covariate adjustment which is shown on page 13 line 35 (Results - Statistical methods for the primary analysis of PROs: paragraph 6).

'There were 106 trials that used multivariable methods, of which (98/106) clearly reported the covariates adjusted in the primary analysis.'

Comment 9: Discussion: Overall the discussion section is balanced, although I'd prefer to see more educational or more elaboration scope in it. The current situation among medical RCTs regarding statistical analyses is suboptimal in many aspects: non-publication bias [14,15], selective reporting [16], secretly HARKing [17], poor conduct and analysis [18], overconfidence about responder analyses [19–22], poor comparability of the results due to endless amount of PRO scales [23–25], and lastly, inadequate understanding of the basic statistics concepts among medical researchers and other readers (clinicians) [26–31] just to name a few examples with overlapping themes with this study. Briefly, I would add to the discussion section some content on obstacles about PROs and statistical analyses in the current atmosphere of medical research.

Response: Thank you for providing these points to enrich the discussion. We have covered publication bias and selective reporting in the study limitation [page 17 line 7-23 (Discussion: paragraph 8)]; and the problem of poor reporting of PRO outcomes is covered on page 18 line 3-10 (Discussion: paragraph 12). We have added a paragraph in the discussion to stress some of these obstacles [page 18 line 26-38 (Discussion: paragraph 14)].

'In addition to the abovementioned issues, other obstacles are also worth attention when analysing PROs in RCT settings, which can be classified into three domains: first, statistics side – inadequate understanding of basic statistical concepts,[78] incorrect procedure to carry out statistical analysis and statistical inference (e.g. secretly hypothesizing after results are known);[79,80] second, PROs side – application of invalid measurements,[81] lack of comparability among the results produced by various PROs[82]; and third, reporting and publishing side – publication bias and selective reporting [82,83]'

References:

- 1 Lumley T, Diehr P, Emerson S, et al. The Importance of the Normality Assumption in Large Public Health Data Sets. *Annu Rev Public Health* 2002;23:151–69. doi:10.1146/annurev.publhealth.23.100901.140546
- 2 Vickers AJ. The use of percentage change from baseline as an outcome in a controlled trial is statistically inefficient: A simulation study. *BMC Med Res Methodol* 2001;1:1–4. doi:10.1186/1471-2288-1-6
- 3 Vickers AJ, Altman DG. Statistics notes: Analysing controlled trials with baseline and follow up measurements. *BMJ* 2001;323:1123–
- 4 <http://www.ncbi.nlm.nih.gov/pubmed/11701584> (accessed 19 Jan 2019).
- 4 Clifton L, Clifton DA. The correlation between baseline score and postintervention score, and its implications for statistical analysis. *Trials* 2019;20:43. doi:10.1186/s13063-018-3108-3
- 5 Ciolino J, Zhao W, Martin R, et al. Quantifying the cost in power of ignoring continuous covariate imbalances in clinical trial randomization. *Contemp Clin Trials* 2011;32:250–9. doi:10.1016/j.cct.2010.11.005
- 6 Ciolino JD, Martin RH, Zhao W, et al. Measuring continuous baseline covariate imbalances in clinical trial data. *Stat Methods Med Res* 2015;24:255–72. doi:10.1177/0962280211416038
- 7 Ciolino JD, Palac HL, Yang A, et al. Ideal vs. real: A systematic review on handling covariates in randomized controlled trials. *BMC Med. Res. Methodol.* 2019;19:1–11. doi:10.1186/s12874-019-0787-8
- 8 Ciolino JD, Martin RH, Zhao W, et al. Covariate imbalance and adjustment for logistic regression analysis of clinical trial data. *J Biopharm Stat* 2013;23:1383–402. doi:10.1080/10543406.2013.834912
- 9 Kahan BC, Morris TP. Reporting and analysis of trials using stratified randomisation in leading medical journals: review and reanalysis. *BMJ* 2012;345:e5840–e5840. doi:10.1136/bmj.e5840
- 10 Kahan BC, Jairath V, Doré CJ, et al. The risks and rewards of covariate adjustment in randomized trials: an assessment of 12 outcomes from 8 studies. *Trials* 2014;15:139. doi:10.1186/1745-6215-15-139
- 11 Hernández A V., Eijkemans MJC, Steyerberg EW. Randomized controlled trials with time-to-event outcomes: How much does prespecified covariate adjustment increase power? *Ann Epidemiol* 2006;16:41–8. doi:10.1016/j.annepidem.2005.09.007

- 12 Hernández A V., Steyerberg EW, Habbema JDF. Covariate adjustment in randomized controlled trials with dichotomous outcomes increases statistical power and reduces sample size requirements. *J Clin Epidemiol* 2004;57:454–60. doi:10.1016/j.jclinepi.2003.09.014
- 13 Saquib N, Saquib J, Ioannidis JPA. Practices and impact of primary outcome adjustment in randomized controlled trials: Meta-epidemiologic study. *BMJ* 2013;347. doi:10.1136/bmj.f4313
- 14 Chan AW, Song F, Vickers A, et al. Increasing value and reducing waste: Addressing inaccessible research. *Lancet*. 2014;383:257–66. doi:10.1016/S01406736(13)62296-5
- 15 Song F, Parekh S, Hooper L, et al. Dissemination and publication of research findings: An updated review of related biases. *Health Technol Assess (Rockv)* 2010;14:1–220. doi:10.3310/hta14080
- 16 Goldacre B, Drysdale H, Dale A, et al. COMPare: A prospective cohort study correcting and monitoring 58 misreported trials in real time. *Trials* 2019;20. doi:10.1186/s13063-019-3173-2
- 17 Hollenbeck JR, Wright PM. Harking, Sharking, and Tharking. *J Manage* 2017;43:5–18. doi:10.1177/0149206316679487
- 18 Ioannidis JPA, Greenland S, Hlatky MA, et al. Increasing value and reducing waste in research design, conduct, and analysis. *Lancet*. 2014;383:166–75. doi:10.1016/S0140-6736(13)62227-8
- 19 Cortés J, González JA, Medina MN, et al. Does evidence support the high expectations placed in precision medicine? A bibliographic review. *F1000Research* 2018;7:30. doi:10.12688/f1000research.13490.5
- 20 Senn S. Seven myths of randomisation in clinical trials. *Stat Med* 2013;32:1439–50. doi:10.1002/sim.5713
- 21 Senn SJ. Statistical pitfalls of personalized medicine. *Nature Published Online First*: 2018. <https://www.nature.com/articles/d41586-018-07535-2>
- 22 Senn S. Individual Therapy: New Dawn or False Dawn? *Drug Inf J* 2001;35:1479–94. doi:10.1177/009286150103500443
- 23 Miyar J, Adams CE. Content and quality of 10 000 controlled trials in schizophrenia over 60 years. *Schizophr Bull* 2013;39:226–9. doi:10.1093/schbul/sbr140
- 24 Wuytack F, Smith V, Clarke M, et al. Towards core outcome set (COS) development: A follow-up descriptive survey of outcomes in Cochrane reviews. *Syst Rev* 2015;4. doi:10.1186/s13643-015-0060-0
- 25 Yordanov Y, Dechartres A, Atal I, et al. Avoidable waste of research related to outcome planning and reporting in clinical trials. *BMC Med* 2018;16:87. doi:10.1186/s12916-018-1083-x
- 26 Localio AR, Stack CB, Meibohm AR, et al. Inappropriate Statistical Analysis and Reporting in Medical Research: Perverse Incentives and Institutional Solutions. *Ann Intern Med* 2018;169:577. doi:10.7326/M18-2516
- 27 Goodman SN. Why is Getting Rid of P -Values So Hard? Musings on Science and Statistics. *Am Stat* 2019;73:26–30. doi:10.1080/00031305.2018.1558111
- 28 Gigerenzer G. Statistical Rituals: The Replication Delusion and How We Got There. *Adv Methods Pract Psychol Sci* 2018;1:198–218. doi:10.1177/2515245918771329
- 29 Gigerenzer G, Gaissmaier W, Kurz-Milcke E, et al. Helping doctors and patients make sense of health statistics. *Psychol Sci Public Interes Suppl* 2007;8:53–96. doi:10.1111/j.1539-6053.2008.00033.x
- 30 Greenland S. Null misinterpretation in statistical testing and its impact on health risk assessment. *Prev Med (Baltim)* 2011;53:225–8.

doi:10.1016/J.YPMED.2011.08.010

31 Greenland S, Senn SJ, Rothman KJ, et al. Statistical tests, P values, confidence intervals, and power: a guide to misinterpretations. Eur J Epidemiol 2016;31:337–50. doi:10.1007/s10654-016-0149-3

Reviewer: 1

Competing interests of Reviewer: none

Reviewer: 2

Competing interests of Reviewer: No

VERSION 2 – REVIEW

REVIEWER	Perrone, Francesco Istituto Nazionale Tumori IRCCS Fondazione Pascale Cura dei , Clinical Trials Unit
REVIEW RETURNED	02-Aug-2021
GENERAL COMMENTS	The Authors answered to all my criticisms and modified the manuscript consistently.